# OpenReview forum: "Adversarial Attacks against Closed-Source MLLMs via Feature Optimal Alignment"
_NeurIPS.cc/2025/Conference — NeurIPS 2025 poster_

### Official Review · Reviewer_swuR · 2025-06-19

**Clarity:** 4
**Significance:** 3
**Originality:** 3
**Rating:** 4
**Confidence:** 3

**Summary:**

This paper proposes a method to produce more transferrable adversarial attacks against closed-source MLLMS. They observe that existing methods produce adversarial examples by minimizing the contrastive loss between the global features (produced by the CLS tokens)  of the example and target samples. They argue that adversarial samples generated this way have poor transferability due to the weak alignment between the local representations (produced by the patch tokens) and the global representations (produced by the CLS token). To solve this problem, the paper proposes FOA-Attack — a method to jointly align global and local features to achieve better transferability. The method uses a clustering technique to distill compact local patterns and frames the alignment of local features as an optimal transport problem. Furthermore, they use an ensemble of models to improve transferability. Experiments show that the proposed method significantly outperforms prior method for generating adversarial examples.

**Questions:**

1. What happens to the attack performance if you perform alignment of local features without using K Means?
2. Any idea why the attack performance on Claude is worse compared to other models?
3. I don’t fully understand Table 6. Do the four rows for FOA have different number of cluster centers?

**Ethical Concerns:**

["NO or VERY MINOR ethics concerns only"]

**Final Justification:**

The authors conducted additional experiments to address my concerns regarding compute-normalized performance comparison and ablation study to understand the impact of k-means clustering. I'm satisfied with the rebuttal and vote to accept the paper.

**Limitations:**

Yes

**Paper Formatting Concerns:**

-

**Quality:**

3

**Strengths And Weaknesses:**

Strengths

- The paper is well written and clearly describes the proposed idea
- The key insight of aligning local features is well motivated
- The results show strong improvement in transferability across multiple target models.

Weaknesses

- The proposed FOA-Attack is significantly more computationally intensive than the primary baseline, M-Attack, taking 217 minutes versus M-Attack's 90 minutes. The paper does not provide a compute-normalized comparison to show how M-Attack would perform if allotted a similar computational budget (e.g., by increasing its attack iterations). Such a comparison would help to more clearly isolate the performance gains attributable to the proposed methodology versus those from simply using more computation.

- The paper argues that clustering is necessary to handle redundant local features, but this is not experimentally validated. The ablation study removes the entire local alignment component but does not include a condition where local alignment is performed on all patch tokens without clustering. This makes it difficult to assess the specific contribution and necessity of the K-Means step.

---

> ### Author Rebuttal · Authors · 2025-07-29
>
> Thank you for your valuable review and suggestions. Below we respond to the comments in **Weaknesses (W)** and **Questions (Q)**.
>
> ---
> ***W1: The proposed FOA-Attack is significantly more computationally intensive than the primary baseline, M-Attack, taking 217 minutes versus M-Attack's 90 minutes. The paper does not provide a compute-normalized comparison to show how M-Attack would perform if allotted a similar computational budget (e.g., by increasing its attack iterations). Such a comparison would help to more clearly isolate the performance gains attributable to the proposed methodology versus those from simply using more computation.***
>
> Thank you for the insightful comment. We agree that compute-normalized comparison is important for a fair evaluation. To address this,  we conducted comparative experiments on 100 randomly selected images. Specifically, we include stronger baselines by increasing M-Attack’s iterations to 700 and 800, which consume 210 and 240 minutes, respectively—comparable to our method (FOA-Attack: 217 min).
>
> As shown in the updated table below, FOA-Attack consistently outperforms both compute-matched baselines across all metrics. Compared to M-Attack (700 iters), FOA-Attack achieves +5%, +7%, and +4% higher ASR on GPT-4o, Claude-3.5, and Gemini, along with consistently higher feature similarity (e.g., 0.62 vs. 0.57 on GPT-4o, 0.18 vs. 0.11 on Claude-3.5). Even against M-Attack (800 iters), which uses slightly more compute, FOA-Attack maintains +3% higher ASR on GPT-4o, +11% on Claude-3.5, and +5% on Gemini, while also surpassing it in feature similarity across all models. These results demonstrate that the improvements of FOA-Attack are due to its methodological design, rather than longer runtime alone.
>
> | Method               | GPT-4o ASR (%) | Avg Sim | Claude-3.5 ASR (%)| Avg Sim | Gemini-2.0 ASR (%) | Avg Sim | Compute Time (min) |
> |---------------------|--------|---------|------------|---------|--------|---------|---------------------|
> | M-Attack            | 73.0   | 0.56    | 10.0       | 0.13    | 48.0      | 0.40       | 90                  |
> | M-Attack (700 iters)| 76.0   | 0.57    | 9.0        | 0.11    | 52.0      | 0.44       | 210                 |
> | M-Attack (800 iters)| 78.0   | 0.59    | 5.0        | 0.09    | 51.0      | 0.43       | 240                 |
> | **FOA-Attack**       | **81.0** | **0.62**  | **16.0**     | **0.18**  |  **56.0**     |  **0.46**    | 217              |
>
> ---
> ***W2 & Q1: The paper argues that clustering is necessary to handle redundant local features, but this is not experimentally validated. The ablation study removes the entire local alignment component but does not include a condition where local alignment is performed on all patch tokens without clustering. This makes it difficult to assess the specific contribution and necessity of the K-Means step. What happens to the attack performance if you perform alignment of local features without using K Means？***
>
>
> Thank you for the valuable question. Following the suggestion, we conduct additional experiments on 100 randomly selected images. As shown in the table below, removing the K-Means clustering step from our local feature alignment module leads to a noticeable drop in attack performance across all evaluated models:
>
> | Method                       | GPT-4o ASR (%) | Avg Sim | Claude-3.5 ASR (%) | Avg Sim | Gemini-2.0 ASR (%) | Avg Sim |
> |-----------------------------|--------|---------|------------|---------|--------|---------|
> | FOA-Attack (w/o KMeans)     | 77.0   | 0.59    | 12.0       | 0.16    | 50.0   | 0.42    |
> | FOA-Attack (w/ KMeans)        | **81.0**   | **0.62**    | **16.0**       | **0.18**    | **56.0**   | **0.46**    |
>
>
> This performance gap confirms the effectiveness of clustering in distilling representative and compact local features, which better capture the semantic structure of the target image. Without K-Means, the alignment operates on all patch tokens, which may include redundant or noisy information and hinder effective optimal transport matching. Therefore, K-Means clustering plays a crucial role in improving both **attack success rate** and **semantic alignment**, especially in the black-box setting. We have now emphasized this finding in our revised version to clarify the importance of this design choice.
>
> ---
> ***Q2:Any idea why the attack performance on Claude is worse compared to other models?***
>
> Thank you for the question. We attribute the lower adversarial attack success rate on Claude to two main factors:
>
> 1. **Robust alignment and safety mechanisms**
>    Claude adopts advanced alignment strategies such as *Constitutional AI*, which are designed to resist harmful or adversarial inputs. For example, Claude-3.5 Sonnet with constitutional classifiers enabled rejects over **95%** of potentially harmful prompts, while the same model without this mechanism only rejects **14%**. These alignment filters are likely also applied to visual prompts or multimodal responses, reducing the likelihood of generating outputs semantically aligned with targeted adversarial images.
>
> 2. **Proprietary vision-language architecture**
>    Unlike open-source models that typically rely on CLIP-style encoders, Claude’s visual processing architecture is undisclosed and possibly features tightly coupled image-text modeling. Prior work[1] has shown that adversarial transferability drops significantly when the target model’s internal representation deviates from that used by the surrogate model. This architectural mismatch makes it more difficult for adversarial perturbations to generalize to Claude.
>
> This observation aligns with recent studies specifically targeting adversarial images. For example, previous work[2] indicates that Claude models are most robust against adversarial attacks.
>
> Hence, Claude’s lower ASR likely stems from a combination of stricter alignment filtering and architectural differences that make adversarial feature alignment less effective. Nevertheless, our **FOA-Attack** still outperforms all baselines on Claude (e.g., **+5.9% ASR over M-Attack on Claude-3.5**), confirming its strong generalization under robust settings.
>
> [1]: Zhao, Yunqing, et al. "On evaluating adversarial robustness of large vision-language models." *NeurIPS*, 2023.
> [2]: Hu, Kai, et al. "Transferable Adversarial Attack on Vision-enabled Large Language Models." (2024).
>
> ---
> ***Q3:I don’t fully understand Table 6. Do the four rows for FOA have different number of cluster centers?***
>
> Thank you for pointing this out. Yes, the four rows for FOA-Attack in Table 6 correspond to different configurations of the number of cluster centers used for local feature alignment. Specifically, we adopt a progressive clustering strategy where the number of cluster centers is increased when the attack fails. The configurations [3], [3,5], [3,5,8], and [3,5,8,10] refer to progressively including more cluster centers during optimization. For example, [3,5] means the attack starts with 3 clusters and increases to 5 if the attack has not yet succeeded. This approach allows our method to better adapt to varying texture complexity across images, improving alignment in both coarse- and fine-grained regions. As shown in Table 6, increasing the number of clusters leads to higher ASR and AvgSim, though at the cost of longer computation time.We have clarified this explanation in the revised version to ensure the setup of Table 6 is more transparent. Thank you for your helpful question.

---

> > ### Comment · Reviewer_swuR · 2025-08-05
> >
> > I thank the authors for providing additional results. This has sufficiently addressed my concerns. I encourage them to include the compute-normalized comparison and ablation results in their paper.

---

> > > ### Author Response · Authors · 2025-08-05
> > > **Official Comment by Authors**
> > >
> > > Dear Reviewer swuR,
> > >
> > > Thank you for your support! We sincerely appreciate your valuable comments and the time you devoted to reviewing our work. Your suggestions have been instrumental in improving the quality of our paper. We are pleased that the additional results have sufficiently addressed your concerns. As recommended, we will incorporate the compute-normalized comparison and ablation results in our next revision. Thank you again for your recognition and encouragement.
> > >
> > > Best, Authors

---

### Official Review · Reviewer_W7EQ · 2025-06-24

**Clarity:** 3
**Significance:** 3
**Originality:** 2
**Rating:** 4
**Confidence:** 3

**Summary:**

The paper proposes a novel approach for generating transferable adversarial examples by maximizing both global and local feature alignment between the adversarial and target examples. For global alignment, the authors employ a feature loss based on the cosine similarity between the [CLS] tokens of the adversarial and target examples. For local alignment, they introduce an optimal transport loss that aligns representative image patch embeddings. These representative patches correspond to cluster centers obtained via k-means clustering applied to all patch embeddings. Additionally, the authors propose an aggregated loss that combines the aforementioned losses across multiple encoders. The weights assigned to each encoder are designed to balance contributions from both fast-learning and slow-learning models.

**Questions:**

* How expensive is the Sinkhorn algorithm in practice? how many iterations is it run for in average until convergence?

* How can such an attack be mitigated?

**Ethical Concerns:**

["NO or VERY MINOR ethics concerns only"]

**Final Justification:**

Well motivated combination of several components that led to significant empirical improvements.

**Limitations:**

The authors include a limitation section.

**Quality:**

3

**Strengths And Weaknesses:**

** Strengths **
* the paper is well written and easy to read.
* the proposed approach is sound and improves upon the baselines


**Weaknesses **
* How expensive is the Sinkhorn algorithm in practice? how many iterations is it run for in average until convergence?
* The approach is incremental (limited novelty)

---

> ### Author Rebuttal · Authors · 2025-07-28
>
> Thank you for your valuable review and suggestions. Below we respond to the comments in **Weaknesses (W)** and **Questions (Q)**.
>
> ---
>
> ***W1&Q1: How expensive is the Sinkhorn algorithm in practice? how many iterations is it run for in average until convergence?***
>
> Thank you for your question. In practice, the Sinkhorn algorithm needs **minimal overhead** compared to the dominant cost of gradient-based optimization in adversarial attacks. Since it only involves matrix-vector multiplications and element-wise operations, its computational cost is negligible relative to backpropagation. Specifically, FOA-Attack takes about 1.13 minutes to generate an adversarial example, whereas the Sinkhorn algorithm accounts for only around 0.015 minutes of that time.
>
> Empirically, the algorithm converges quickly. On average, it runs for **22 iterations per image** during each attack iteration, with the following breakdown across surrogate models:
>
> - **ViT-B/16**: 7 iterations
> - **ViT-B/32**: 7 iterations
> - **ViT-g/14 (Laion)**: 8 iterations
>
> These results confirm that the Sinkhorn-based optimal transport alignment is both **efficient and practical** for integration in the inner loop of adversarial optimization.
>
> ---
> ***W2: The approach is incremental (limited novelty).***
>
> We respectfully disagree with the assessment that our novelty is limited. As one of the first works to enhance adversarial transferability for closed-source MLLMs, our method systematically addresses key limitations in prior methods like M-Attack and AttackVLM. Specifically, we go beyond global-only alignment and fixed ensemble losses by introducing novel components that substantially improve attack generalizability:
> - **Joint Global-Local Feature Alignment:**  While existing methods primarily focus on global features (e.g., the [CLS] token), our method is the first to explicitly combine both coarse-grained and fine-grained feature alignment by formulating local feature matching as an optimal transport (OT) problem (Section 3.3). We introduce a local clustering OT loss, which significantly boosts semantic consistency and adversarial transferability, especially toward closed-source commercial MLLMs, which prior works struggle to attack effectively (Tables 2, 3, and 5).
> - **Dynamic Ensemble Weighting Strategy:**  Unlike prior ensemble methods that treat all surrogate models equally (e.g., fixed weights in M-Attack), we propose a learning-speed-based dynamic loss weighting strategy (Section 3.4, Eq. 13). This technique adaptively down-weights easier-to-optimize objectives to prevent overfitting to specific models, resulting in better transferability to unseen black-box targets. To our knowledge, such convergence-aware balancing has not been applied to adversarial optimization for MLLMs before.
> -  **Progressive Clustering for Transfer-Efficiency Tradeoff:**  We introduce a progressive local clustering scheme where the number of local cluster centers is increased only if the attack fails, balancing attack success and computational efficiency (Table 6). This dynamic strategy improves transferability while controlling resource cost, which is crucial for scalable adversarial evaluations.
> -  **Superior Performance Across Diverse Models:** Our proposed method achieves state-of-the-art ASR and AvgSim on 14 models, including GPT-4o, Claude-3.7, and Gemini-2.0, significantly outperforming strong baselines such as M-Attack (Tables 1, 2, 3). This demonstrates not only empirical superiority but also the practical significance of our novel techniques.
>
> In summary, **FOA-Attack** offers conceptual, algorithmic, and empirical innovations that go beyond incremental improvements. We hope the reviewer will reconsider the novelty and impact of our contributions in light of these advancements.
>
> ---
> ***Q2: How can such an attack be mitigated?***
>
> Thank you for the valuable question. Such attacks can be mitigated from two complementary perspectives:
>
> 1. **Pre-processing Purification**: One approach is to eliminate adversarial perturbations before inference. For example, Pei et al. [1] propose a diffusion-based purification method from the frequency domain, which evolves adversarial noise through the forward and reverse processes, effectively removing harmful perturbations while preserving semantic content.
>
> 2. **Robust Encoder Training**: Another line of defense involves enhancing the adversarial robustness of the image encoder itself. Hossain and Imteaj [2] introduce an adversarial training scheme to secure vision-language models against both jailbreak and adversarial attacks by injecting adversarial examples into the encoder training process.
>
>
> [1] Pei, Gaozheng, et al. *"Diffusion-based Adversarial Purification from the Perspective of the Frequency Domain."* ICML, 2025.
> [2] Hossain, Md Zarif, and Ahmed Imteaj. *"Securing vision-language models with a robust encoder against jailbreak and adversarial attacks."* IEEE BigData, 2024.

---

> > ### Author Response · Authors · 2025-08-05
> > **Rebuttal follow-up**
> >
> > Dear Reviewer W7EQ,
> >
> > As the discussion period progresses, we would like to kindly inquire whether our response has sufficiently addressed your concerns. We remain at your disposal for any further clarifications and greatly appreciate your valuable feedback.
> >
> > Best, Authors

---

> ### Comment · Reviewer_W7EQ · 2025-08-06
>
> Thank you! My concerns were addressed.

---

> > ### Author Response · Authors · 2025-08-06
> > **Official Comment by Authors**
> >
> > Dear Reviewer W7EQ,
> >
> > Thank you very much for your kind feedback. We are glad to hear that your concerns have been addressed. Your comments and suggestions have been extremely valuable in helping us improve the quality and clarity of our work. We sincerely appreciate the time and effort you dedicated to reviewing our paper.
> >
> > Best, Authors

---

### Official Review · Reviewer_9n7D · 2025-07-02

**Clarity:** 4
**Significance:** 3
**Originality:** 3
**Rating:** 5
**Confidence:** 4

**Summary:**

FOA-Attack is a novel targeted adversarial attack framework for MLLMs that enhances transferability by aligning both global and local features in the image encoder. It utilizes a global cosine similarity loss and a local clustering optimal transport loss for coarse-grained and fine-grained feature alignment, respectively. The experiments demonstrate higher ASR on open-sourced and closed-sourced models like GPT, Gemini, and Claude compared to existing targeted attacks.

**Questions:**

See weaknesses

**Ethical Concerns:**

["NO or VERY MINOR ethics concerns only"]

**Final Justification:**

All of my concerns were addressed.

I don't think this paper is groundbreaking (this is an established setting and the technical advancements over prior work are not extremely novel) and therefore I didn't give it a 6. However, all weaknesses are addressed and I strongly recommend accepting this paper.

**Limitations:**

addressed

**Quality:**

4

**Strengths And Weaknesses:**

## Strengths
- The methodology is sound, using an optimal transport based objective to align local features is elegant and novel
- The empirical results are very compelling. I like that the authors focus on evaluating ASR on closed sourced models.


## Weaknesses
- The paper does not consider using targets that may elicit unsafe or harmful captions from the target model. This setting is underexplored, but poses a more real-world risk.
- Both the source and target images are natural images, but it is unclear how similar do the source and target images need to be for attacks to be successful.
- The set of image encoders in the ensemble are all CLIP models. Increasing the diversity of the ensemble may further improve performance. For example, what happens if we mix CLIP and SigLIP models in the ensemble?
- [Minor] Below Line 180, the optimization problem should have $min_{\pi}$ instead of just $min$ for clarity.

---

> ### Author Rebuttal · Authors · 2025-07-29
>
> Thank you for your valuable review and suggestions. Below we respond to the comments in **Weaknesses (W)** and **Questions (Q)**.
>
> ---
> ***W1: The paper does not consider using targets that may elicit unsafe or harmful captions from the target model. This setting is underexplored, but poses a more real-world risk.***
>
> Thank you for the insightful suggestion. To evaluate the attack performance in more realistic and safety-critical scenarios, we conducted additional experiments using **harmful target images** designed to elicit unsafe or harmful captions. Specifically,
> we conduct experiments on 100 randomly selected source images. Then, we randomly selected **100 harmful images** as the target images, covering **five categories**: *Harassment*, *Violence*, *Self-harm*, *Sexual*, and *Illegal activity*, with 20 images per category. These images were sampled from the benchmark proposed in [1].
>
> We then applied our adversarial attack methods (M-Attack and FOA) to generate adversarial samples from these target images. Each target model was instructed to describe the adversarially perturbed images. To determine whether the output was indeed harmful, we used GPT-4o as an automatic safety classifier.
>
> The results, shown in the table below, demonstrate that our method (FOA) consistently leads to higher rates of unsafe generation across all categories and models compared to the baseline (M-Attack), especially under high-risk categories such as *Illegal activity* and *Self-harm*. This highlights the importance of considering harmful target prompts in safety evaluations.
>
> | Model   | Method   | Harassment | Violence | Self-harm | Sexual | Illegal activity | Avg  |
> |---------|----------|------------|----------|-----------|--------|------------------|------|
> | GPT-4o  | M-Attack | 30.0       | 40.0     | 25.0      | 10.0   | 50.0             | 31.0 |
> |         | FOA-Attack     | **35.0**   | **45.0** | **40.0**  | **15.0** | **50.0**         | **37.0** |
> | Claude  | M-Attack | 5.0        | 0.0      | 5.0       | 0.0    | 5.0              | 3.0  |
> |         | FOA-Attack      | **10.0**   | **5.0**  | **10.0**  | **5.0**| **10.0**         | **8.0** |
> | Gemini  | M-Attack | 35.0       | 30.0     | 25.0      | 10.0   | 40.0             | 28.0 |
> |         | FOA-Attack      | **45.0**   | **30.0** | **30.0**  | **20.0** | **45.0**         | **34.0** |
>
> These findings underscore the potential real-world risks when models are exposed to adversarial samples derived from inherently harmful content, further validating the threat model considered in our paper.
>
> [1] Qu, Yiting, et al. "Unsafebench: Benchmarking image safety classifiers on real-world and ai-generated images." 2024.
>
> ---
> ***W2: Both the source and target images are natural images, but it is unclear how similar do the source and target images need to be for attacks to be successful.***
>
>
> Thank you for the thoughtful question. We compare CLIP similarities between successful and failed samples across various models. The results are shown below:
>
> | CLIP Similarity | Qwen2.5-VL-3B | Qwen2.5-VL-7B | LLaVa-1.5-7B | LLaVa-1.6-7B | Gemma-3-4B | Gemma-3-12B | Claude-3.5 | Claude-3.7 | GPT-4o | GPT-4.1 | Gemini-2.0 |
> |----------------------|------------------|------------------|------------------|------------------|------------------|-------------------|----------------|--------------------|----------------|----------------|------------------|
> | Success         | 0.4719           | 0.4742           | 0.4714           | 0.4711           | 0.4742           | 0.4767            | 0.4890         | 0.4762             | 0.4713         | 0.4714         | 0.4739           |
> | Fail             | 0.4656           | 0.4560           | 0.4588           | 0.4607           | 0.4656           | 0.4646            | 0.4662         | 0.4674             | 0.4610         | 0.4600         | 0.4604           |
>
> On average, successful samples exhibit slightly higher CLIP similarity (~+0.009), suggesting that visual similarity between source and target images can slightly benefit attack success. However, the gap is small, indicating that our method does not rely heavily on similarity and remains robust across diverse natural images.
>
> ---
>
> ***W3: The set of image encoders in the ensemble are all CLIP models. Increasing the diversity of the ensemble may further improve performance. For example, what happens if we mix CLIP and SigLIP models in the ensemble?.***
>
>
> We appreciate the reviewer’s insightful suggestion. To evaluate the impact of ensemble diversity, we conducted additional experiments on 100 randomly selected images by mixing CLIP and non-CLIP models such as SigLIP into the ensemble. The results are shown in the table below:
>
> | Method                         | GPT-4o ASR (%) | AvgSim | Claude-3.5 ASR (%) | AvgSim | Gemini ASR (%) | AvgSim |
> |--------------------------------|----------------|--------|---------------------|--------|----------------|--------|
> | M-Attack                       | 73.0           | 0.56   | 10.0                | 0.13   | 48.0           | 0.40   |
> | FOA-Attack                     | 81.0           | 0.62   | 16.0                | 0.18   | 56.0           | 0.46   |
> | M-Attack (+SigLIP)             | 73.0           | 0.56   | 11.0                | 0.13   | 50.0           | 0.41   |
> | FOA-Attack (+SigLIP)           | 82.0           | 0.62   | 19.0                | 0.20      | 60.0           | 0.50   |
>
> Adding SigLIP leads to consistent improvements across all target models when applied to our method. Specifically, FOA-Attack (+SigLIP) achieves 82.0% ASR on GPT-4o, 19.0% on Claude-3.5, and 60.0% on Gemini, all higher than the original FOA-Attack setting. These results clearly indicate that introducing encoder diversity via SigLIP can enhance adversarial transferability, especially to closed-source models. Moreover, compared to M-Attack (+SigLIP), our FOA-Attack (+SigLIP) achieves significantly better transfer performance across all targets. This indicate the effectiveness of our method’s core components—local-global feature alignment and dynamic ensemble weighting—in harnessing the added diversity from SigLIP. It also demonstrates that FOA-Attack is better equipped to adapt to heterogeneous surrogate encoders, achieving both higher ASR and semantic consistency.
>
>
>
>
> ---
>
> ***W4: Below Line 180, the optimization problem should have $ \min_{\pi} $ instead of just $ \min $ for clarity.***
>
>
> Thank you for the suggestion. We agree with the reviewer that adding the subscript π improves clarity. We will revise the notation from $ \min $ to $ \min_{\pi} $ below Line 180 in the final version.

---

> > ### Author Response · Authors · 2025-08-05
> > **Rebuttal follow-up**
> >
> > Dear Reviewer 9n7D,
> >
> > As the discussion period progresses, we would like to kindly inquire whether our response has sufficiently addressed your concerns. We remain at your disposal for any further clarifications and greatly appreciate your valuable feedback.
> >
> > Best, Authors

---

> ### Comment · Reviewer_9n7D · 2025-08-05
> **Response to Rebuttal**
>
> Thank you for the excellent rebuttal.
>
> All of my questions are addressed, and I hope the authors include the new results in the updated manuscript. I keep my score, and recommend acceptance.

---

> > ### Author Response · Authors · 2025-08-05
> > **Official Comment by Authors**
> >
> > Dear Reviewer 9n7D,
> >
> > Thank you very much for your positive feedback and kind recommendation. We are glad to hear that our rebuttal has addressed your questions. As suggested, we will include the new results in the updated manuscript. We sincerely appreciate your support and recommendation for acceptance.
> >
> > Best, Authors

---

### Official Review · Reviewer_NA4a · 2025-07-03

**Clarity:** 2
**Significance:** 3
**Originality:** 3
**Rating:** 4
**Confidence:** 3

**Summary:**

This paper proposes a new adversarial attack method against closed source MLLMs, which not only use global features but also adopt local features to align the adversarial sample with target sample.

**Questions:**

1.	The method use clustering technique to clustering the local features, how to select the cluster number. As the number of clusters also affect the attack performance.

2.	The generation of the final adversarial sample is not given, how do you obtain the adversarial sample based on the global and local alignment?

3.	How many models are used in Eq.(11). And are they only CLIP models? Why not use more different models to improve the transferability.

**Ethical Concerns:**

["NO or VERY MINOR ethics concerns only"]

**Final Justification:**

Thanks for the authors' reply. I will raise the score.

**Limitations:**

Yes

**Paper Formatting Concerns:**

NAN

**Quality:**

3

**Strengths And Weaknesses:**

Strengths:
1.	This paper proposes a targeted transferable adversarial attack method based on both global and local feature-alignment.

2.	Clustering techniques and a local feature alignment method, i.e., a local clustering OT are employed, due to the redundant local features of the image.

3.	A dynamic ensemble model weighting strategy is proposed for improving the adversarial transferability, which is based on the models’ convergence rates.

Weaknesses:

1.	The method use clustering technique to clustering the local features, how to select the cluster number. As the number of clusters also affect the attack performance.

2.	The generation of the final adversarial sample is not given, how do you obtain the adversarial sample based on the global and local alignment?

3.	How many models are used in Eq.(11). And are they only CLIP models? Why not use more different models to improve the transferability.

---

> ### Author Rebuttal · Authors · 2025-07-29
>
> Thank you for your valuable review and suggestions. Below we respond to the comments in **Weaknesses (W)** and **Questions (Q)**.
>
> ---
>
> ***W1 & Q1: The method uses clustering technique to clustering the local features, how to select the cluster number. As the number of clusters also affect the attack performance.***
>
> Thank you for the insightful question. As you correctly pointed out, the number of cluster centers is indeed a crucial factor influencing the effectiveness of the attack. We conduct experiments with different numbers of cluster centers on 100 randomly selected images.  As shown in the table below, different numbers of clusters lead to varying levels of adversarial transferability. This is primarily because each image possesses unique texture patterns, resulting in different optimal numbers of clusters for their local features.
>
> To address this variability, we adopt a progressive clustering strategy, as detailed in Lines 193–195 of our manuscript. Specifically, we begin with a small number of clusters and incrementally increase it when the attack fails. This adaptive mechanism enables our method to better accommodate diverse texture distributions across images.
>
> To systematically investigate the impact of the number of clusters, we conduct an ablation study in Section 4.5, with results presented in Table 6. The experiments show that increasing the number of cluster centers generally improves attack success rates and feature alignment metrics. However, this improvement comes with additional computational overhead. To strike a balance between effectiveness and efficiency, we adopt the [3,5] setting (i.e., progressively increasing from 3 to 5 cluster centers) in our main experiments. This configuration achieves a favorable trade-off across different datasets and attack scenarios.
>
> We have now made this design choice and its motivation clearer in the revised version to avoid potential confusion. Thank you again for your valuable feedback.
>
>
> | Method                   | Claude-3.5 ASR (%)  | Claude-3.5 AvgSim | GPT-4o ASR (%)  | GPT-4o AvgSim | Gemini ASR (%)  | Gemini AvgSim |
> |--------------------------|----------------|-------------------|------------|----------------|------------|----------------|
> | M-Attack                 | 10.0           | 0.13              | 73.0       | 0.56           | 48.0       | 0.40           |
> | FOA-Attack ([3])         | 12.0           | 0.14              | 76.0       | 0.58           | 53.0       | 0.44           |
> | FOA-Attack ([5])         | 10.0            | 0.13              | 75.0       | 0.58           | 52.0       | 0.44           |
> | FOA-Attack ([8])         | 11.0           | 0.14              | 73.0       | 0.57           | 51.0       | 0.42           |
> | FOA-Attack ([10])        | 10.0           | 0.13              | 71.0       | 0.56           | 51.0       | 0.43           |
> | FOA-Attack ([3,5])       | 16.0           | 0.18              | 81.0       | 0.62           | 56.0       | 0.46           |
> | FOA-Attack ([3,5,8])     | 17.0           | 0.20              | 83.0       | 0.63           | 59.0       | 0.48           |
> | FOA-Attack ([3,5,8,10])  | 18.0           | 0.21              | 84.0       | 0.64           | 61.0       | 0.50           |
>
> ---
> ***W2 & Q2: The generation of the final adversarial sample is not given, how do you obtain the adversarial sample based on the global and local alignment?***
>
> Thank you for your question.  We do not need the final adversarial sample to calculate the loss — we only compute the loss between the **current** adversarial sample and the target image at each iteration to guide the update. Specially, as shown in **Algorithm 1** of the supplementary material, we initialize the adversarial sample by adding a small perturbation to the clean image. Then, we iteratively update it by minimizing a joint loss that aligns **global features** (via cosine similarity of [CLS] tokens) and **local features** (via Optimal Transport between clustered patch features). The final gradient is aggregated across surrogate models with dynamically adjusted weights based on their convergence speed, and used to update the perturbation under an ℓ∞ constraint.
>
> ---
> ***W3 & Q3: How many models are used in Eq.(11). And are they only CLIP models? Why not use more different models to improve the transferability.***
>
> Thank you for your valuable question. In Eq. (11), we use three CLIP-based image encoders, consistent with the setup adopted in the original M-Attack paper, which also ensembles three CLIP variants. We retain this setting in our main experiments to ensure a fair and direct comparison.
>
> In response to your suggestion, we additionally explore the effect of incorporating more diverse models into the ensemble. Specifically, we include SigLIP and MetaCLIP to construct a stronger ensemble. We conduct experiments on 100 randomly selected images.  The results are presented in the table below:
>
> | Method                         | GPT-4o ASR (%) | AvgSim | Claude-3.5 ASR (%) | AvgSim | Gemini ASR (%) | AvgSim |
> |--------------------------------|----------------|--------|---------------------|--------|----------------|--------|
> | M-Attack                       | 73.0           | 0.56   | 10.0                | 0.13   | 48.0           | 0.40   |
> | FOA-Attack                     | 81.0           | 0.62   | 16.0                | 0.18   | 56.0           | 0.46   |
> | M-Attack (+SigLIP)             | 73.0           | 0.56   | 11.0                | 0.13   | 50.0           | 0.41   |
> | FOA-Attack (+SigLIP)           | 82.0           | 0.62   | 19.0                | 0.20      | 60.0           | 0.50   |
> | M-Attack (+MetaCLIP)             | 74.0           | 0.58   | 13.0                | 0.15   | 51.0           | 0.42   |
> | FOA-Attack (+MetaCLIP)           | 85.0           | 0.64   | 24.0                | 0.24      | 60.0           | 0.51   |
> | M-Attack (+SigLIP+MetaCLIP)    | 75.0           | 0.58  | 13.0                 | 0.15   | 53.0           | 0.43   |
> | FOA-Attack (+SigLIP+MetaCLIP)  | **86.0**       | **0.64** | **26.0**           | **0.25** | **62.0**     | **0.53** |
>
> As seen, expanding the ensemble to include more diverse models (SigLIP, MetaCLIP) improves transferability across all target models. Notably, our proposed **FOA-Attack consistently outperforms M-Attack in all settings**, confirming that the effectiveness of FOA-Attack is not solely dependent on the backbone ensemble size or diversity, but benefits from our proposed optimization strategies.

---

> > ### Author Response · Authors · 2025-08-05
> > **Rebuttal follow-up**
> >
> > Dear Reviewer NA4a,
> >
> > As the discussion period progresses, we would like to kindly inquire whether our response has sufficiently addressed your concerns. We remain at your disposal for any further clarifications and greatly appreciate your valuable feedback.
> >
> > Best, Authors

---

> > > ### Comment · Reviewer_NA4a · 2025-08-08
> > >
> > > Thank you for your feedback. I have raised the rating score.

---

> > > > ### Author Response · Authors · 2025-08-08
> > > > **Official Comment by Authors**
> > > >
> > > > Dear Reviewer NA4a,
> > > >
> > > > Thank you very much for your kind feedback and for raising the rating score. We truly appreciate your recognition and support. Your comments have been invaluable in improving our work, and we will carefully incorporate the suggested enhancements in the revised manuscript.
> > > >
> > > > Best, Authors

---

### Note · Authors · 2025-08-12

We thank all reviewers and area chairs for their constructive feedback and engagement. Our rebuttal addressed the concerns, leading to rating improvements and recommendations for acceptance.

**Rebuttal Highlights:**
1. **Cluster Number & Progressive Strategy:** Clarified adaptive clustering, supported by ablations (Table 6), balancing transferability and efficiency.
2. **Adversarial Sample Generation:** Detailed iterative joint optimization of global (cosine) and local (OT) feature alignment with dynamic ensemble weighting.
3. **Ensemble Diversity:** Added SigLIP/MetaCLIP experiments, showing consistent gains and robustness beyond backbone choice.
4. **Safety-Critical Scenarios:** Included harmful target images (5 categories), confirming FOA-Attack’s real-world risk relevance.
5. **Compute Fairness:** Matched M-Attack’s runtime via extended iterations; FOA-Attack still outperformed across all targets.
6. **Necessity of Clustering:** New ablations showed notable drops without K-Means, confirming its role in removing redundancy.
7. **Claude Gap Analysis:** Attributed to strong alignment filters and architectural differences.
8. **OT Efficiency:** Sinkhorn adds negligible overhead (<1.5 s per example) and converges quickly.

**Key Contributions:**
- **Joint Global–Local Feature Alignment:** First to integrate coarse- and fine-grained alignment in MLLM attacks, formulating local alignment as an OT problem with clustering.
- **Dynamic Ensemble Weighting:** Convergence-aware balancing to enhance transferability to unseen black-box models.
- **Progressive Clustering:** Adaptively increases cluster centers only when needed for success-efficiency tradeoff.
- **Extensive Evaluation:** Achieves SOTA ASR/semantic similarity on 14 models, including GPT-4o, Claude, and Gemini, under both standard and harmful-content settings.
- **Practical Impact:** Reveals critical vulnerabilities of commercial MLLMs, guiding future safety assessments and defenses.

We appreciate the reviewers’ insights, which have helped strengthen the technical rigor, clarity, and real-world relevance of our work.

---

### Decision · Program_Chairs · 2025-09-17

**Decision:**

Accept (poster)

**Comment:**

This paper proposes FOA-Attack, which is a targeted transferable adversarial attack method for closed-source multimodal LLMs that jointly aligns global and local features through cosine similarity and optimal transport with clustering with the technique of dynamic ensemble weighting.\
Reviewers initially raised concerns about clustering choices, computational cost, and novelty. The rebuttal provided thorough clarifications, including ablations on clustering, compute-normalized baselines, ensemble diversity (SigLIP/MetaCLIP), and evaluation on harmful-content settings. Rebuttals addressed the main criticisms and strengthened the work, and then consensus among reviewers was reached. It is encouraged that authors include those revisions in the final draft.\
Overall, this paper makes a decent contribution to understanding and improving adversarial transferability for commercial MLLMs.